# Supporting people with type 2 diabetes in effective use of their medicine through mobile health technology integrated with clinical care (SuMMiT-D Feasibility): a randomised feasibility trial protocol

Andrew Farmer [1], Julie Allen,[1] Kiera Bartlett,[2] Peter Bower,[3] Yuan Chi,[4] David French [2], Bernard Gudgin,[5] Emily A Holmes,[6] Robert Horne,[7] Dyfrig A Hughes,[8] Cassandra Kenning [9], Louise Locock,[10] Jenny McSharry,[11] Lisa Miles,[12] Nikki Newhouse [ ], Rustam Rea,[1] Evgenia Riga,[1] Lionel Tarassenko,[4] Carmelo Velardo,[4] Nicola Williams,[1] Veronika Williams,[1] Ly-Mee Yu,[1] On behalf of the SuMMiT-D Collaborative Group

For numbered affiliations see end of article.

**Correspondence to**
Professor Andrew Farmer;
andrew.farmer@phc.ox.ac.uk

## ABSTRACT

**Introduction** Type 2 diabetes is common, affecting over 400 million people worldwide. Risk of serious complications can be reduced through use of effective treatments and active self-management. However, people are often concerned about starting new medicines and face difficulties in taking them regularly. Use of brief messages to provide education and support self-management, delivered through mobile phone-based text messages, can be an effective tool for some long-term conditions. We have developed messages aiming to support patients' self-management of type 2 diabetes in the use of medications and other aspects of self-management, underpinned by theory and evidence. The aim of this trial is to determine the feasibility of a large-scale clinical trial to test the effectiveness and cost-effectiveness of the intervention, compared with usual care.

**Methods and analysis** The feasibility trial will be a multicentre individually randomised, controlled trial in primary care recruiting adults (≥35 years) with type 2 diabetes in England. Consenting participants will be randomised to receive short text messages three times a week with messages designed to produce change in medication adherence or non-health-related messages for 6 months. The aims are to test recruitment methods, retention to the study, the feasibility of data collection and the mobile phone and web-based processes of a proposed definitive trial and to refine the text messaging intervention. The primary outcome is the rate of recruitment to randomisation of participants to the trial. Data, including patient reported measures, will be collected online at baseline and the end of the 6-month follow-up period. With 200 participants (100 in each group), this trial is powered to estimate 80% follow-up within 95% CIs of 73.8% to 85.3%. The analysis will follow a prespecified plan.

### Strengths and limitations of this study

► The use of brief messages to support self-management and provide motivation and education, delivered through mobile phone-based text messages, can be an effective tool for some long-term conditions.

► There are a few trials testing the impact of brief messaging in type 2 diabetes; although some have systematically developed messages mapped to theoretical constructs, many are at risk of bias and may be limited in their application to specific contexts.

► The objective of the SuMMiT-D Feasibility trial is to test recruitment and randomisation of participants and collection of proposed primary and secondary outcome data in planning for a large, outcomes based, randomised controlled trial.

► The messages evaluated in this study are: (1) targeted at constructs identified as being related to medication adherence by systematic reviews of the literature and (2) demonstrated to have fidelity to intended behaviour change techniques (as rated by experts) and acceptability to target population (as rated by patients).

► Potential mechanisms for the action of the brief messages will be assessed through changes in constructs that are related to medication-taking behaviour, questions based on the technology acceptance model and self-report medication adherence.

**Ethics and dissemination** Ethics approval was obtained from the West of Scotland Research Ethics Committee 05. The results will be disseminated through conference presentations, peer-reviewed journals and will be published on the trial website: www.summit-d.

org (SuMMiT-D (SUpport through Mobile Messaging and digital health Technology for Diabetes)).

**Trial registration number** ISRCTN13404264.

## INTRODUCTION

Type 2 diabetes is one of the most common long-term conditions affecting 422 million people worldwide[1] and 4.7 million people in the UK.[2] It can lead to major complications including cardiovascular disease, renal failure and neuropathy.[3] The global burden of diabetes is projected to reach up to 2.2% of global gross domestic product,[4] and many of these costs are due to preventable complications. Prevention includes the use of treatments of proven efficacy[5 6] alongside supporting self-management.[7] However, concerns about medicines and difficulties in taking them regularly, whether intentional or unintentional, are common[8] and lead to poor diabetes control and increased use of health resources.[9] The cost of non-adherence to diabetes medication in the UK has been estimated at £100 million per year in avoidable treatment costs.[10] A wide range of services are available to support people in better use of their medicines, but evidence of their effectiveness and cost-effectiveness is weak.[11] These services are often targeted at particular groups of individuals and are frequently designed as 'one-off' interventions. Examples include medication reviews, education and addressing cost issues.[12] Understanding and improving this situation could make a major contribution to health and healthcare costs.

Systematic reviews do not provide strong evidence to support the wider use of current approaches to developing interventions to improve adherence.[13] Brief messages delivered at a wide scale and low cost via digital health systems added to usual care have been shown to be effective in improving health for some conditions and are a promising approach to the problem.[14]

Systematic reviews of text messages to support patients to adhere to treatment, and of mobile health interventions in diabetes, identify some effective interventions. There are a few trials testing the impact of brief messaging in type 2 diabetes; although some have systematically developed messages mapped to theoretical constructs, many are at risk of bias and may be limited in their application to specific contexts.[14 15] Recent trials of text messaging for cardiovascular risk prevention and blood pressure lowering have shown clinically relevant changes in outcomes compared with usual care.[16 17]

In addition, there is substantial evidence that tailored interventions are more effective than generic interventions.[18] Tailored interventions may be seen by recipients as more personally relevant, so they will be more likely to attend to, read, understand and act on them. In addition, tailored interventions are designed to change determinants of the target behaviour that are relevant to particular individuals or to small subgroups of individuals; they therefore more precisely target the determinants of the individual's behaviour.

SUpport through Mobile Messaging and digital health Technology for Diabetes (SuMMiT-D) is a programme of work composed of three phases: formative work, a feasibility trial and a large scale, effectiveness randomised controlled trial of a mobile phone-based system intended to deliver brief, tailored, behaviour change messages to people with type 2 diabetes focusing on use of medication. The intervention is intended to focus on a broad range of individuals with type 2 diabetes, but those with younger onset diabetes and using insulin alone were not included, as these features can require care using different pathways. In the formative work for this trial, we identified theoretical constructs and features of intervention content found to be associated with medication adherence in patients with type 2 diabetes and mapped these onto a standard taxonomy for behaviour change techniques (BCTs), that is, active ingredients of interventions used to promote behavioural change.[19 20] We then developed a large set of messages to target each BCT and examined which types of messages are most useful and easy to understand for people starting and taking an oral diabetes medicine and the extent to which it might be helpful for patients to decide on the types of messages they want to receive. We received input from approximately 300 patients with type 2 diabetes and healthcare professionals caring for this patient population. Preliminary findings of this trial development work were used to develop this feasibility trial with scope for further refinement of the system for the main clinical trial.

The primary objective of the SuMMiT-D Feasibility trial is to test recruitment and randomisation of participants to the trial. We will test collection of planned primary and secondary outcome data in planning for a large, outcomes-based, randomised controlled trial. We will assess the feasibility and acceptability of the intervention for patients and healthcare professionals; the willingness of participants to be randomised; follow-up rates; resource use; and trial procedures. We will also carry out a process evaluation of how the system is used and refine the way it is embedded within usual care. Potential mechanisms for the action of the brief messages will be assessed through changes in hypothesised health psychology constructs relating to use of medication, self-report medication adherence,[21] and questions based on the technology acceptance model.[22]

## METHODS AND ANALYSIS

The SuMMiT-D Feasibility trial protocol is reported according to Standard Protocol Items: Recommendations for Interventional Trials (SPIRIT) and Medication Adherence Reporting Guidelines (EMERGE) recommendations.[23 24]

### Patient and public involvement (PPI)

Patient members of the public are integral to this trial. A panel of 11 PPI members with type 2 diabetes was set up and continues to inform our work, reviewing all patient

documentation and research findings, and supports the development of the intervention.

All patient facing documents for the SuMMiT-D Feasibility trial, including the participant information sheet, informed consent form, posters, user guides and website, were reviewed by PPI panel members. The panel has been kept up to date with frequent trial updates in the form of an email newsletter, also published on the trial website ( www.summit-d.org). The results of the study will be made available to trial participants, PPI panel members and participating genenral practices practices on the trial website.

## Trial design

SuMMiT-D Feasibility is a primary care-based, two-arm, individually randomised controlled, parallel group trial aiming to recruit a total of 200 patients across 20 general practice sites in England. Patients with type 2 diabetes will be randomly allocated to receive an individually tailored short messaging system (SMS) text messaging-based intervention for 26 weeks that aims to encourage and support them in developing a habit of taking their medication as intended (ie, to promote effective implementation of dosing and treatment continuation)[25] and provides hints and tips to help them with other aspects of living with

the condition alongside usual care (treatment arm) or to usual care with the addition of infrequent non-health-related messages (control arm) (figure 1).

## Intervention

### Intervention arm: condition-specific tailored text messaging system and usual care

Participants assigned to the intervention group will receive brief health-related SMS text messages, based on systematic review of the evidence identifying determinants of medication-taking behaviour.[20] Messages were developed based on systematic review evidence by experts[20] and were refined in an iterative process of ensuring acceptability based on patient feedback and demonstration of fidelity to intended behaviour change determinants, as rated by an independent group of experts.[26] A more detailed description of the intervention is given in the accompanying template for intervention description and replication (TIDieR) checklist. Examples of messages are given in the supplementary file (see online supplementary appendix 1).

The intervention is a digital health system with the following components:

i. Participants will be sent up to four automated text messages per week with an average frequency of

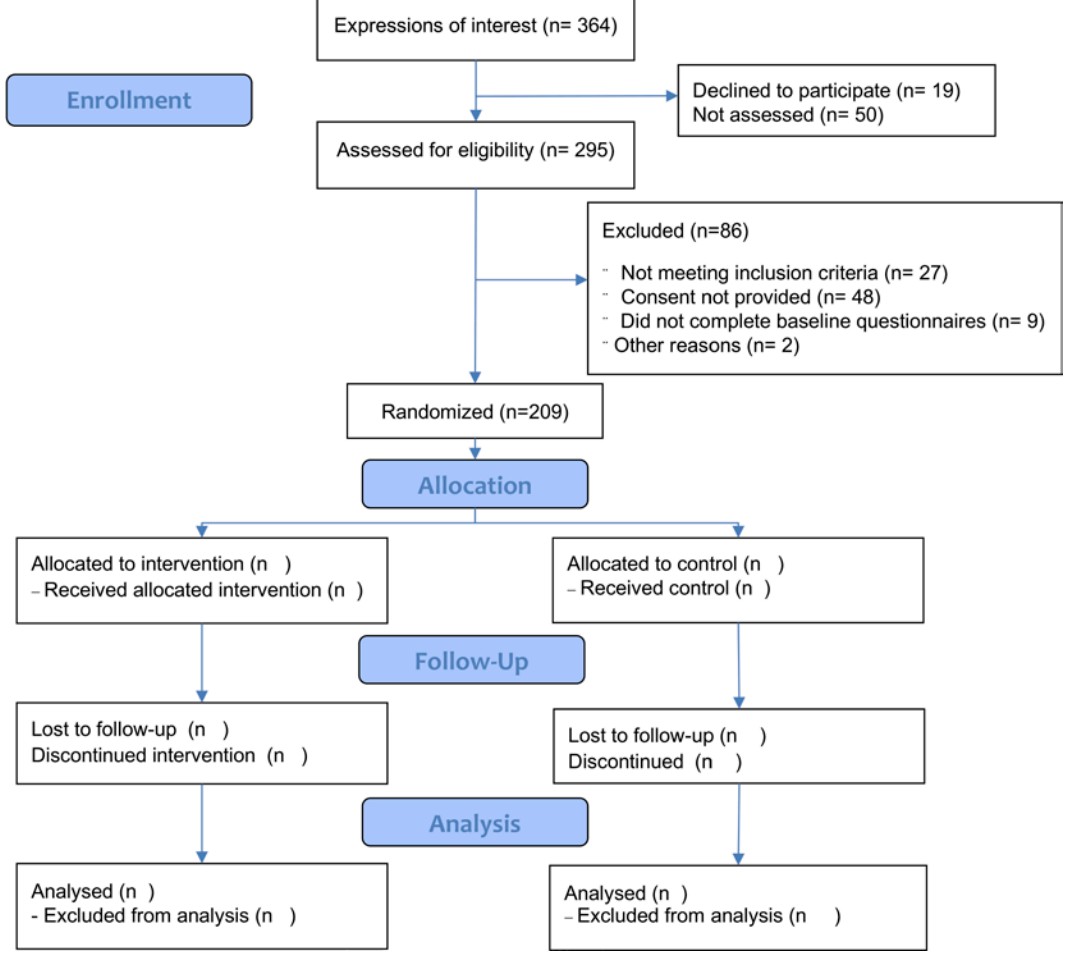

**Figure 1** Trial profile.

three per week relating to diabetes management and use of medicine.

ii. The library of text messages uses different BCTs to target health-related behaviour change relating to use of medicines, as well as messages targeting other aspects of diabetes care (including diet and exercise).

iii. The types of messages sent will be tailored to participants based on time since starting new glucose, blood pressure and cholesterol-lowering medication and current smoking status.

iv. Frequency of messages received using a particular group of BCTs can be modified based on a participant's response to individual messages received through sending a text message in response to a particular message. Participants may incur a cost for sending messages in response depending on their network plan.

v. The style of messages will be patient centred and will encourage patients to seek further relevant information (including the use of links where possible to selected external websites, eg, Diabetes UK).

### Control arm: usual care and one non-health-related message a month

Participants assigned to usual care will receive non-health-related text messages at a frequency of approximately one every 4 weeks. Care will otherwise not differ from usual care.

### Outcomes

The primary outcome is the rate of recruitment to randomisation of participants to the trial. Details of outcomes and measurement are shown in table 1. We will measure recruitment against planned recruitment rates for the proposed main trial and number of people showing an interest in the trial and not proceeding or those who withdraw from the control group and give a reason.

Secondary outcomes (table 1) focus on feasibility of collection of clinical and economic measurement data for the proposed main trial. They include the availability of HbA1c, systolic blood pressure and total to HDL cholesterol ratio data from medical records, retention rates and proportion of prescribing data available. A self-report questionnaire will assess: medication adherence,[21] health status with the EuroQol 5-Dimension, 5-Level (EQ-5D-5L),[27] resource use for the health economic analysis, technology acceptance[22] and constructs targeted by SMS and hypothesised to mediate the effects of the intervention on adherence.[28]

### Procedures and assessments

Participants who express interest in taking part in the trial will be screened by the trial team and will consent and submit their baseline questionnaires either online or on paper according to their preference. Participants will be randomised by the trial team and will receive messages for 26 weeks from randomisation to final follow-up. All participants will be asked to complete the same set of questionnaires at baseline and at the end of their 26-week follow-up period. Medical notes reviews will be conducted at baseline, 6 months and 18 months from randomisation.

### Recruitment

Potential participants will be identified through general practices in England. Participating practices fall under four Clinical Research Networks: Thames Valley & South Midlands, Greater Manchester, West Midlands and South West Peninsula.

Healthcare professionals will screen their type 2 diabetes clinic lists and will invite eligible patients. Patients will receive an invitation letter and summary information leaflet. Eligible patients may also be contacted by phone, email or text by the practice team up to three times.

### Expressions of interest

People interested in taking part can send their full name by SMS text message to the trial team to register their interest. If potential participants have any difficulties in registering their interest in the trial, they will also be able to contact the team via phone or email and will receive support in registering as required.

### Screening assessment

Following an expression of interest, further information about the study will be provided by email or by post as requested. Eligibility will be confirmed by phone.

### Inclusion criteria

Eligible participants are ≥35 years of age, taking oral glucose lowering treatment, blood pressure-lowering treatment or lipid-lowering treatment either alone or in combination. They have access to a mobile phone and are able, if necessary, with help (eg, relative, friend, neighbour), to send, understand and retrieve brief SMS text messages in the English language. Participants who are using insulin treatment without also concomitant use of oral glucose-lowering treatment; who are pregnant, within 3 months postpartum or planning pregnancy during the course of the trial; have a serious medical condition that, in the opinion of the investigator, makes them ineligible; have been admitted to hospital within the last 3 months for hyperglycaemia or hypoglycaemia, are ineligible.

### Informed consent

Participants will provide consent either online or on paper.

### Baseline and follow-up assessments

The following questionnaires will be administered (online or by post) at the baseline assessment: the Medication Adherence Report Scale (MARS) Self-Report Scale,[21] the EQ-5D-5L Scale,[27] a set of measures developed for the study based on the technology acceptance model[22] and a healthcare utilisation record to allow healthcare resource use to be costed. A further set of measures assess the constructs that will be targeted by the BCTs within the messages. These constructs are hypothesised to mediate

**Table 1**  Schedule of trial outcomes and measures

| Outcome | Measure | Timing (months) | | |
| --- | --- | --- | --- | --- |
| | | 0 | 6 | 18 |
| Participant recruitment to the trial. | Recruitment against planned recruitment rates. | × | | |
| Participant willingness to be randomised. | Number of people showing an interest and not proceeding or those who withdraw from the control group and give a reason. | × | | |
| Feasibility of collection of clinical measurement data for the proposed clinical trial. | Retrieval of measurements of HbA1c, systolic blood pressure and cholesterol for trial participants from medical record. | × | × | |
| Willingness of participants to be followed up over the 26-week period postrandomisation. | Retention and follow-up rates. | | × | |
| Feasibility of collection of prescribing data on trial participants. | Proportion of medication possession ratio for glucose, blood pressure and lipid-lowering medication obtainable from prescribing data in participant's medical record. | | × | |
| Feasibility of collecting self-reported questionnaire measures. | Proportion of completed self-reported measures (Demographics and Additional Information Questionnaire; MARS Self-Report Scale; EQ-5D-5L Health Status Questionnaire; Healthcare Utilisation Record Questionnaire (cost measurement); and Health Psychology and Technology Acceptance Questionnaire. | × | × | |
| Feasibility of collecting medical history and baseline medication from medical record. | Proportion of data obtained. | × | | |
| Feasibility and acceptability of the intervention for patients and healthcare professionals (including general practitioners, nurses, receptionists and pharmacists). Qualitative process evaluation. | Data obtained through focus groups, qualitative interviews with patients and healthcare staff. | × | × | |
| Assess reliability of measures of hypothesised mechanism of action and sensitivity to change in response to receipt of SMS messages. Examine relationship between these measures and self-reported adherence, as preliminary process analysis. (Quantitative process evaluation). | Change in quantitative mechanism of action measures and relationship between changes in these measures and self-reported adherence. | × | × | |
| Changes in clinical measurement data. | HbA1c, systolic blood pressure and cholesterol for trial participants from medical record. | × | × | × |
| Information on message delivery and interaction with participants. | Automated reports from messaging service on messages delivered and interactive messaging. | × | × | |

EQ-5D-5L, EuroQol 5-Dimension, 5-Level; HbA1c, glycated haemoglobin; MARS, Medication Adherence Report Scale.

the effects of the intervention on medication taking behaviour, in line with the logic model we have developed based on the Health Action Process Approach (figure 2, table 1 and online supplementary appendix 2).[28]

All self-report data will be collected either online or on paper according to the participant's preference. The same measures are collected at 26 weeks after randomisation alongside clinical record data.

### Randomisation

Participants will be randomised after consent and when all baseline assessments have been completed. Participants will be allocated in a 1:1 ratio to either the intervention or the control arm. Randomisation will be done using a validated secure web-based randomisation programme (Sortition) provided by the University of Oxford Primary Care Clinical Trials Unit (PC-CTU). Allocation will be carried out with a non-deterministic minimisation algorithm to ensure groups are balanced for important baseline prognostic and other factors: study site, age (<65/≥65 years), gender (male/female), duration of diabetes (<5 years/≥5 years) and number of medications (<5/≥5). The allocated intervention will be implemented directly by the platform on which the digital health system is run. Apart from the qualitative research team and the engineering team, allocation is blinded to all other trial and healthcare staff. Due to the nature of the study, unblinding is not required during the trial.

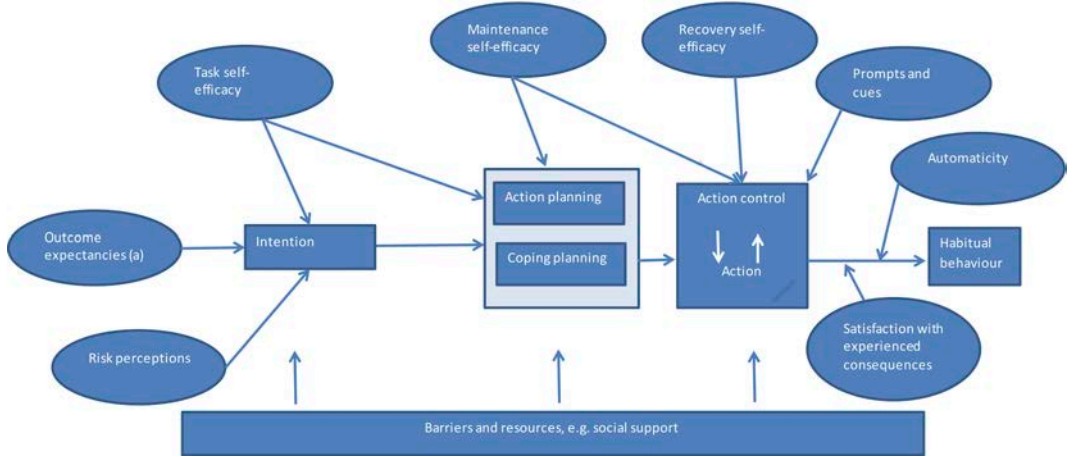

**Figure 2** Logic model showing anticipated mechanisms of action in the SuMMIT-D Feasibility study, based on the Health Action Process Approach. (a) This construct includes both beliefs about the necessity of taking medication, and concerns about taking that medication.

### Discontinuation of intervention or withdrawal from trial

Participants can withdraw from the trial at any time. Participants can also choose to pause or stop receipt of text messages by sending a text message or contacting the trial office by telephone or post. Adverse events are collected, and serious unexpected adverse events related to the intervention are determined by the chief investigator and reported in line with local procedures.

### Statistical analysis
#### Power

With 200 participants (100 in each group), the feasibility trial is powered to estimate 80% follow-up within 95% CIs of 73.8% to 85.3%.

#### Analysis

The primary outcome will be the number of patients recruited to randomisation as a proportion (with 95% CI) of the target recruitment number. Secondary outcomes will be reported overall and separately by allocated arm.

An intention-to-treat approach will be used for an exploratory analysis of secondary outcomes by allocated arm. Continuous outcomes will be analysed using an analysis of covariance adjusting for minimisation factors. Results will be presented as an adjusted difference in means with 95% CI. Binary outcomes will be analysed in a similar way using log binomial regression models (adjusting for minimisation factors). Results will be presented as relative risks with 95% CIs. There will be no formal assessment of treatment efficacy for this feasibility study.

### Analysis of utility and costs

Data collected on healthcare utilisation will be assessed for completeness. The items recorded include patient contacts with healthcare professionals, use of primary care services, hospital outpatient appointments, accident and emergency visits, and inpatient stays, prescribed and purchased medications, and personal social services. The proportion of complete and returned questionnaires will be reported. Total costs associated with each item of resource will be estimated based on unit costs from standard sources[29 30] and presented descriptively.

The costing is conducted from the perspective of the National Health Service (NHS), and therefore the cost to the patient is not considered; we will, however, estimate costs of responding to text messages using estimates from average network tariffs and data collected on the quantity of digital information transmitted.

Responses to the EQ-5D-5L questionnaire will also be examined for completeness and reported by proportion of returned questionnaires and items with a valid response. Health utility values corresponding to patients' EQ-5D-5L profiles will be calculated by mapping the 5L descriptive system data onto the 3L value set[31] and presented descriptively.

### Process evaluation

We will perform a mixed-methods process analysis, consisting of a qualitative element and a quantitative element. The qualitative process evaluation will examine the experience of participants and health professionals in their use of and implementation of the messaging system. Longitudinal interviews with up to 30 participants will take prior to use of the system and after the 26-week follow-up. Additionally, up to 30 healthcare professionals will be invited to share their experiences of taking part in this trial.

#### Interviews with trial participants

Participants who consent to taking part in the embedded qualitative study will be purposefully selected by characteristics including age, gender, length of time of diabetes and type of medication and invited to share their views on engagement with and content of the messaging system and to provide insight into how it was implemented in daily life, identify issues around potential attrition and inform final procedures for the main trial.

### Interviews and/or focus groups with healthcare professionals

Healthcare staff from practices taking part in the trial will be invited to take part in focus groups or qualitative interviews to share their experience of recruiting for this trial and how the intervention was implemented in routine clinical care. Eligible staff will be those with a potential role in the implementation of a text message system or who have contact with patients with type 2 diabetes. These will include GPs, practice nurses, receptionists and healthcare assistants.

### Analysis of qualitative data: trial participants

Thematic analysis will be used inductively to identify key issues, focussing on similarities and differences identified in themes to understand the issues that are important to the participants.[32] We will use NVivo and mind-mapping software to code data, identify themes and interpret these and their relationship. The data analysis will be inductive.

### Analysis of qualitative data: healthcare professionals

We will follow a semistructured topic guide based on the core constructs of extended normalisation process theory (NPT)[33]: coherence, reflexive monitoring, cognitive participation and collective action to ensure key areas are covered while allowing flexibility to follow up topics introduced by participants. NVivo software will be used to facilitate data organisation, analysis and the development of themes. The analysis will use a thematic approach informed by NPT.[32]

### Quantitative process analysis

To facilitate understanding of mechanism of the intervention, the effect of the intervention on the set of measures based on the logic model (figure 2 and table 1) will be analysed using a per-protocol analysis for responders, according to allocated group. This will identify if these measures have been affected by the intervention. A further analysis will examine how these measures in the logic model (controlling for baseline) predict change in self-reported medication adherence using hierarchical multiple regression techniques.[34] This analysis will thereby assess extent of association between measures in the logic model and self-reported medication adherence.[35] These analyses will be used in conjunction with the qualitative analyses described above to inform decisions about selection of SMS messages relating to particular BCTs and potentially refinement of the logic model and selection of measures for the full trial. Receipt of messages on participants' mobile phones will be monitored throughout the trial.

### Data management

All trial data will be entered on electronic case report forms. The clinical database is built on Research Electronic Data Capture, a secure, web-based application designed to support data capture for research studies.[36]

### Ethics and dissemination

The trial will be conducted according to the principles of the Declaration of Helsinki and in accordance with other relevant national guidelines, regulations, acts and using Good Clinical Practice guidelines. The University of Oxford sponsors the trial.

The role of the Trial Steering Committee is taken on by the National Institute for Health Research Programme Steering Committee.

### Dissemination plan

The results of this trial will be submitted to a peer-reviewed journal for publication.

Recruitment to the SuMMiT-D Feasibility study began with the first participant randomised on 26 November 2018 and the last participant randomised on 16 April 2019. Reporting of the trial is anticipated in the first quarter of 2020.

### DISCUSSION

SuMMiT-D Feasibility will inform the design of a future large-scale randomised controlled trial, aiming to estimate the clinical and cost-effectiveness of the text messaging intervention and identify difficulties that might be encountered in practice. If effective and implemented in the NHS, this intervention could help reduce the burden of complications and increased costs associated with non-adherence. This research could also offer a model for technology-based self-management support that could be extended to other aspects of diabetes care and other long-term conditions.

The SuMMiT-D Collaborative Study Group are listed in online supplementary appendix 3 with their roles.

**Author affiliations**
[1]Nuffield Department of Primary Care Health Sciences, University of Oxford, Oxford, UK
[2]The Division of Psychology and Mental Health, The University of Manchester, Manchester, UK
[3]Division of Population Health, Health Services Research & Primary Care, The University of Manchester, Manchester, UK
[4]The Institute of Biomedical Engineering, University of Oxford, Oxford, UK
[5]Patient Representative, Oxford, UK
[6]School of Health Sciences, Bangor University, Bangor, UK
[7]UCL School of Pharmacy, University College London, London, UK
[8]Centre for Health Economics and Medicines Evaluation, Bangor University, Bangor, UK
[9]Institute of Population Health, The University of Manchester, Manchester, UK
[10]Health Service Research, University of Aberdeen, Aberdeen, UK
[11]Health Behaviour Change Research Group, School of Psychology, National University of Ireland Galway, Galway, Ireland
[12]Division of Psychology and Mental Health, The University of Manchester, Manchester, UK

**Acknowledgements** The SUpport through Mobile Messaging and digital health Technology for Diabetes research team acknowledge the support of the National Institute for Health Research (NIHR) through the Clinical Research Networks (AF:

 

NIHR Senior Investigator and AF and RR: funding from NIHR Oxford Biomedical Research Consortium).

**Contributors** Each author has contributed significantly to and is willing to take public responsibility for, one or more aspects of the study. AF, LL and DF conceived the study; AF, DF, PB, LL, RR, VW, LL, JM, CV, LT, DAH, RH and LM-Y were involved in planning the study. all authors are involved in carrying out the study. KB, YC, EAH, CK, LM, NN and ER are involved in data collection. AF and ER drafted the initial manuscript; all authors provided revisions and approved the final version.

**Funding** This article presents independent research funded by the NIHR under its Programme Grants for Applied Research as part of a wider programme of work (RP-PG-1214-20003).

**Disclaimer** The views expressed are those of the authors and not necessarily those of the NHS, the NIHR or the Department of Health and Social Care.

**Competing interests** LT reported personal fees from Sensyne Health, CV reported salary support from Sensyne Health and RH reported grants and personal fees from AstraZeneca, personal fees from GSK, personal fees from TEVA, personal fees from Amgen, personal fees from Astellas, personal fees from Abbvie, personal fees from Novartis, from Allergan, outside the submitted work and is a director of Spoonful of Sugar Ltd, a University College London Business providing consultancy on treatment engagement and patient support programmes to healthcare policy makers, providers and industry.

**Patient consent for publication** Not required.

**Provenance and peer review** Not commissioned; externally peer reviewed.

**ORCID iDs**
Andrew Farmer http://orcid.org/0000-0002-6170-4402
David French http://orcid.org/0000-0002-7663-7804
Cassandra Kenning http://orcid.org/0000-0001-6041-4051
Nikki Newhouse http://orcid.org/0000-0001-9002-3588

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
