## [Reviewer comments · BMJ Open]

ARTICLE DETAILS

TITLE (PROVISIONAL)	Supporting people with type 2 diabetes in effective use of their medicine through mobile health technology integrated with clinical care (SuMMiT-D Feasibility): a randomised feasibility trial protocol
AUTHORS	Farmer, Andrew; Allen, Julie; Bartlett, Kiera; Bower, Peter; Chi, Yuan; French, David; Gudgin, Bernard; Holmes, EA; Horne, Robert; Hughes, Dyfrig; Kenning, Cassandra; Locock, Louise; McSharry, Jenny; Miles, Lisa; Newhouse, Nikki; Rea, Rustam; Riga, Evgenia; Tarassenko, Lionel; Velardo, Carmelo; Williams, Nicola; Williams, Veronika; Yu, Ly-Mee

VERSION 1 - REVIEW

REVIEWER	NW CHeung Westmead Hospital NSW Australia
REVIEW RETURNED	21-Sep-2019

GENERAL COMMENTS	This is a protocol paper for the SuMMiT-D Feasibility trial, a feasibility trial for a larger RCT aiming to test the effectiveness of text messaging to improve medication adherence amongst people with T2 diabetes. The trial has already completed recruitment, with its primary aim to assess the rate of recruitment to the trial. The requisite SPIRIT checklist is in order and the protocol itself appears sound. Questions and comments. In the introduction there is discussion that there is a lack of evidence for services to support medication adherence, and current approaches to developing interventions to improve adherence. It would be helpful if the investigators can list a few examples of these interventions and approaches. The primary outcome is to assess the rate of recruitment to the trial, which is stated to be 200 participants in 3 months. The recruitment reported indicates that 209 subjects have been randomised over a period of 5 months. Does this mean that the trial has already failed to meet its primary outcome? This may need to be clarified. It would be helpful for more details of the secondary outcomes to be included in the paper. The way it is written, "completeness of
---

	HbA1c ...” implies that the investigators are assessing whether these tests have been performed, which is more an evaluation of the healthcare provider rather than the participant’s behaviour. I am unclear what is meant by “proportion of expected medication use data available”. It appears from the protocol that medical records will be reviewed at baseline and after 6 months to assess medications prescribed. This is slightly expanded on in the protocol, but I am still not clear if the “proportion of that expected” really just means proportion for whom there is data available, as we should expect 100%. Could the investigators please elaborate on this.
--	---

REVIEWER	Rosie Dobson University of Auckland, New Zealand
REVIEW RETURNED	01-Oct-2019

GENERAL COMMENTS	The authors should be congratulated on a well-designed study. This paper presents a study protocol for a randomised controlled trial of a text message behaviour change programme for people with type 2 diabetes. They should be commended on their robust methodology in designing and developing their intervention and proposed methods for assessing its feasibility. Although the paper is worthy of publication there are a number of clarifications that should be made to ensure the manuscript adequately describes the study. Additionally the layout of the methods section could be improved to aid comprehension. My specific comments below are listed according to section. Abstract:  1. Please state the study primary outcome in the abstract. Introduction:  1. “There are a few trials testing the impact of brief messaging in type 2 diabetes, but they do not have systematically-developed interventions based on theory and evidence and are at risk of bias.” I don’t believe this statement is completely correct, although I absolutely agree that most previous messaging programmes for diabetes have not been systematically developed nor are theoretically based this is not correct for all. Recent publication of studies that are evidence based and systematically developed mean that the authors blanket statement is no longer valid. For example our study published last year in the BMJ was both theoretically based (targeting BCTs) and systematically developed. Although designed for both type 1 and type 2 diabetes it is individually tailored and findings are reported by type. I am not expecting the authors to reference our paper but it is just an example of why i think it might be good to rephrase the above statement.  2. As per above I feel the authors statement in the article summary also needs updating: “There are a few trials testing the impact of brief messaging in type 2 diabetes, but they do not have systematically developed interventions targeting a range of behaviour change techniques.” Methods and analysis:
--

	1. I think the layout of this section could be improved to increase comprehensibility. Perhaps a table summarising the different outcomes and associated measures and time points of measurement would be useful. Not all outcomes and measures are described in the processdures, outcomes and measures sections. Patient and public involvement: 2. I would recommend involving your PPI members in the interpretation of the results as well as this can be extremely valuable. Intervention; Intervention arm. 3. I would recommend changing he first line to “Participants assigned to the intervention group...” rather than “participants assigned to usual care...” to avoid confusion with the control group 4. More detail on the structure and content of the programme is needed, I.e. different types of messages, topics and examples of messages. 5. You state that messages will be tailored, is this based on baseline characteristics or is the programme tailored throughout the duration of the programme? 6. In point #iv. You mention “participant response to individual messages” this implies that it is a 2-way programme. I think that this needs to be described further. Are all messages 2-way? Is there a feedback mechanism built into the programme for responses? What happens if they don’t reply? Does it cost to reply? 7. Are there any costs to the participant to receive this programme (send or receive messages)? Outcomes: 8. The outcomes section should include a complete list of secondary measures and how they are being measures (self-report, patient records etc.). This section does not describe the interviews (“process evaluation”) and does not mention where data such as “medication use” will be obtained from (I had to go to the main protocol document to get this). The statement “Data on changes in clinical parameters will be reported” needs further detail/explanation. 9. I would recommend using a consistent structure throughout when describing the different measures and outcomes i.e. health economic measures, process evaluation 10. Is engagement with the intervention being measured? If so how? Procedures and assessments; Inclusion criteria 11. I think the authors need to justify why they have excluded people under the age of 35 when the prevalence of diabetes in younger adults is increasing worldwide and this group are more likely to live with the condition for a longer period and therefore have higher risks of developing debilitating complications of the condition. 12. The introduction to the protocol describes type 2 diabetes generally but the inclusion criteria is very specific and the authors are excluding many people with this diagnosis in the study e.g. those on insulin. I therefore think it is important that the rationale for this is clearly explained in this section or the introduction is updated to focus on the target group of the intervention.
--	--

	13. As well as justifying why those on insulin are excluded it would be good to justify why those who have been hospitalised for hyper or hypoglycaemia are excluded as you could argue that this group would most benefit from this type of intervention considering it targets medication management Procedures and assessments; Baseline and follow up assessments 14. “A further set of measures based on the logic model...” Which measures are these? Are these described in the outcomes section? Are these measures the ones described later in the section “Quantitative process analysis”? I would recommend further detail here on what is being measured, how many items and in what format.
--	---

VERSION 1 – AUTHOR RESPONSE

Comment	Response	
Reviewer: 1		
This is a protocol paper for the SuMMiT-D Feasibility trial, a feasibility trial for a larger RCT aiming to test the effectiveness of text messaging to improve medication adherence amongst people with T2 diabetes. The trial has already completed recruitment, with its primary aim to assess the rate of recruitment to the trial. The requisite SPIRIT checklist is in order and the protocol itself appears sound.	Thank you for this comment	
In the introduction there is discussion that there is a lack of evidence for services to support medication adherence, and current approaches to developing interventions to improve adherence. It would be helpful if the investigators can list a few examples of these interventions and approaches.	We have added references to the Academy of Sciences report on adherence and to a brief review published in JAMA citing “...examples including medication reviews, education and addressing cost issues.”	✓
The primary outcome is to assess the rate of recruitment to the trial, which is stated to be 200 participants in 3 months. The recruitment reported indicates that 209 subjects have been randomised over a period of 5 months. Does this mean that the trial has already failed to meet its primary outcome? This may need to be clarified.	The phrase “We aim to recruit 200 people in three months” was extracted from operational documents prepared after the protocol had been submitted and the trial registered. We have therefore removed it from the manuscript. The following sentence “We will measure recruitment against planned recruitment rates for the proposed main trial” was the original phrase in the protocol and on the trial registration website.	✓
It would be helpful for more details of the secondary outcomes to be included in the paper. The way it is written, “completeness of HbA1c ...” implies that the investigators are assessing whether these tests have been performed, which is	We have now included and referenced a table (Table 1) giving more details about the secondary outcomes and their timing. We have rephrased the section on “completeness” to clarify that we are establishing whether this data can be	✓

more an evaluation of the healthcare provider rather than the participant's behaviour.	collected from the medical record on participants.	
I am unclear what is meant by "proportion of expected medication use data available". It appears from the protocol that medical records will be reviewed at baseline and after 6 months to assess medications prescribed. This is slightly expanded on in the protocol, but I am still not clear if the "proportion of that expected" really just means proportion for whom there is data available, as we should expect 100%. Could the investigators please elaborate on this.	Thank you for pointing this out. This refers to medication prescribing data availability. We have rephrased. It now reads "proportion of prescribing data available ". The new table of outcome measures clarifies that we will be using this data to calculate a medication possession ratio for each participant	✓
Reviewer: 2 Rosie Dobson		
The authors should be congratulated on a well-designed study. This paper presents a study protocol for a randomised controlled trial of a text message behaviour change programme for people with type 2 diabetes. They should be commended on their robust methodology in designing and developing their intervention and proposed methods for assessing its feasibility. Although the paper is worthy of publication there are a number of clarifications that should be made to ensure the manuscript adequately describes the study. Additionally the layout of the methods section could be improved to aid comprehension. My specific comments below are listed according to section.	Thank you – the comments below are helpful and we have addressed them fully.	
Abstract:		
1. Please state the study primary outcome in the abstract.	We have added the following sentence "The primary outcome is the rate of recruitment to randomisation of participants to the trial."	✓
Introduction:		

1. "There are a few trials testing the impact of brief messaging in type 2 diabetes, but they do not have systematically-developed interventions based on theory and evidence and are at risk of bias." I don't believe this statement is completely correct, although I absolutely agree that most previous messaging programmes for diabetes have not been systematically developed nor are theoretically based this is not correct for all. Recent publication of studies that are evidence based and systematically developed mean that the authors blanket statement is no longer valid. For example our study published last year in the BMJ was both theoretically based (targeting BCTs) and systematically developed. Although designed for both type 1 and type 2 diabetes it is individually tailored and findings are reported by type. I am not expecting the authors to reference our paper but it is just an example of why i think it might be good to rephrase the above statement.	Apologies – the phrase slipped though from the original grant application and we recognise that other work has taken this agenda forward whilst our programme of work has been progressing. We have rephrased to ensure that we do not make an inaccurate statement. The text now says "There are a few trials testing the impact of brief messaging in type 2 diabetes, although some have systematically-developed messages mapped to theoretical constructs, many are at risk of bias and may be limited in their application to specific contexts.^{12,13}"	✓
2. As per above I feel the authors statement in the article summary also needs updating: "There are a few trials testing the impact of brief messaging in type 2 diabetes, but they do not have systematically developed interventions targeting a range of behaviour change techniques."	We have rephrased as above	✓
Methods and analysis:		
1. I think the layout of this section could be improved to increase comprehensibility. Perhaps a table summarising the different outcomes and associated measures and time points of measurement would be useful. Not all outcomes and measures are described in the procedures, outcomes and measures sections.	We have added a table as suggested and confirm that this matches the primary and secondary outcome data listed on our trial registration.	✓
Patient and public involvement:		
2. I would recommend involving your PPI members in the interpretation of the results as well as this can be extremely valuable.	This section was not well worded and implied the PPI members had only helped in the formative work. This is incorrect and they are a key part of our work. We have rephrased "continues to inform our work, reviewing all patient documentation and research findings,...":	✓
Intervention; Intervention arm.		

3. I would recommend changing the first line to “Participants assigned to the intervention group...” rather than “participants assigned to usual care...” to avoid confusion with the control group	Thank you for this helpful suggestion. We have phrased as suggested	✓
4. More detail on the structure and content of the programme is needed, i.e. different types of messages, topics and examples of messages.	We apologise for not including reference to the accompanying TIDieR document that contains this detail. We have also added examples of types of messages in the on-line supplement.	✓
5. You state that messages will be tailored, is this based on baseline characteristics or is the programme tailored throughout the duration of the programme?	Messages are tailored in a number of ways. Participants can send texts to change the time of receiving messages, they can pause or stop messages and they can send a “like” or dislike” to individual messages. This flag is used to increase or decrease the number of messages sent from a particular group of messages.	✓
6. In point #iv. You mention “participant response to individual messages” this implies that it is a 2-way programme. I think that this needs to be described further. Are all messages 2-way? Is there a feedback mechanism built into the programme for responses? What happens if they don't reply? Does it cost to reply?	See point 4 and 5 above. If the participant does not respond, then they will continue to receive messages selected at the pre-determined frequency from across the groups of messages. The majority of participants do not incur costs from replying to messages (but see following point). “...participant's response to individual messages received through sending a text-message in response to a particular message. Participants may incur a cost for sending messages in response depending on their network plan.”	✓
7. Are there any costs to the participant to receive this programme (send or receive messages)?	Participants incur no costs in receiving messages. If they respond, then they may incur the costs of sending a text-message. However, we know from our formative work that the great majority of participants have contracts that allow them to send messages without cost. We have addressed this in the trial by offering a gift voucher of a value greater than any possible cost to them of sending text messages as a token of thanks for participating. In terms of the health economic analysis The microcosting is conducted from the perspective of the NHS and therefore the cost to the patient is not considered, we will however, estimate costs of responding to text messages using estimates from average network tariffs	✓

	and data collected on the quantity of digital information transmitted.	
Outcomes:		
8. The outcomes section should include a complete list of secondary measures and how they are being measured (self-report, patient records etc.). This section does not describe the interviews (“process evaluation”) and does not mention where data such as “medication use” will be obtained from (I had to go to the main protocol document to get this). The statement “Data on changes in clinical parameters will be reported” needs further detail/explanation.	We have now included a detailed table (Table 1) listing outcomes, measure and timing. The new table now mentions the additional outcomes associated with the process evaluation. The process evaluation has a section later in the protocol. The prescribing data in the medical record is now specified as the source of data for the medication possession ratio, The phrase “Data on changes in clinical parameters will be reported” has been changed to “An exploratory analysis will be carried out on HbA1c, systolic blood pressure and cholesterol to explore changes in over time.”	✓
9. I would recommend using a consistent structure throughout when describing the different measures and outcomes i.e. health economic measures, process evaluation	We have included reference to the different components of the evaluation in the new table. We have also avoided using the term “health economic” in favour of “utility” and “cost” measurement	✓
10. Is engagement with the intervention being measured? If so how?	Engagement is monitored in the qualitative interviews and by monitoring the receipt of messages. We have included brief statements to specify this (page 11.12 and 13 of the manuscript).	✓
Procedures and assessments; Inclusion criteria		
11. I think the authors need to justify why they have excluded people under the age of 35 when the prevalence of diabetes in younger adults is increasing worldwide and this group are more likely to live with the condition for a longer period and therefore have higher risks of developing debilitating complications of the condition.	We recognise the increasing prevalence of type 2 diabetes in the younger population. However, in non-high risk groups, diagnosis at a younger age is less common and can follow a differing treatment pathway to rule out atypical type 1 diabetes or other atypical conditions. The intervention was designed to engage a wide group of people with type 2 diabetes but was not intended to be used by a younger population or those using insulin alone who might, in any case, be receiving a different care pathway. We have clarified the text. “The intervention is intended to focus on a broad range of individuals with type 2 diabetes, but those with younger onset diabetes and using insulin alone were not included as	✓

	these features can require care using different pathways.”	
12. The introduction to the protocol describes type 2 diabetes generally but the inclusion criteria is very specific and the authors are excluding many people with this diagnosis in the study e.g. those on insulin. I therefore think it is important that the rationale for this is clearly explained in this section or the introduction is updated to focus on the target group of the intervention.	The introduction has been amended as outlined in response to the previous point.	✓
13. As well as justifying why those on insulin are excluded it would be good to justify why those who have been hospitalised for hyper or hypoglycaemia are excluded as you could argue that this group would most benefit from this type of intervention considering it targets medication management	We have not excluded those treated with insulin from the trial, only those on insulin alone. The reason for this, and also excluding those with hyper or hypoglycaemia is that those individuals who experience these complications are likely to be receiving additional support and care. We do not think that the type of messages developed to support medication adherence or general diabetes care are appropriate for these individuals who may be experiencing specific difficulties that would not be appropriately managed by brief messages.	✓
Procedures and assessments; Baseline and follow up assessments		
14. “A further set of measures based on the logic model...” Which measures are these? Are these described in the outcomes section? Are these measures the ones described later in the section “Quantitative process analysis”? I would recommend further detail here on what is being measured, how many items and in what format.	Thank you. The new table (Table 1) refers to the health beliefs and behaviour measures and their timing. We have updated this section to refer to the table and have also added the relevant measure as a supplementary file to provide full details of items and format.	✓

VERSION 2 – REVIEW

REVIEWER	N Wah Cheung Westmead Hospital Australia
REVIEW RETURNED	17-Nov-2019

GENERAL COMMENTS	Only one minor comment, that the dissemination plan is in the manuscript twice, under both Ethics and Dissemination, and Data. This should not be necessary.
--

REVIEWER	Rosie Dobson University of Auckland, New Zealand
REVIEW RETURNED	18-Nov-2019

GENERAL COMMENTS	The authors have adequately addressed my concerns and questions from my previous review, thank you.
---